# Comparison of Various Cell Lines and Three-Dimensional Mucociliary Tissue Model Systems to Estimate Drug Permeability Using an In Vitro Transport Study to Predict Nasal Drug Absorption in Rats

**DOI:** 10.3390/pharmaceutics12010079

**Published:** 2020-01-17

**Authors:** Tomoyuki Furubayashi, Daisuke Inoue, Noriko Nishiyama, Akiko Tanaka, Reiko Yutani, Shunsuke Kimura, Hidemasa Katsumi, Akira Yamamoto, Toshiyasu Sakane

**Affiliations:** 1Department of Pharmaceutical Technology, Kobe Pharmaceutical University, 4-19-1 Motoyamakitamachi, Higashinada-ku, Kobe 658-8558, Japan; a-tanaka@kobepharma-u.ac.jp (A.T.); r-yutani@kobepharma-u.ac.jp (R.Y.); 2School of Pharmacy, Shujitsu University, 1-6-1 Nishigawara, Naka-ku, Okayama 703-8516, Japan; d-inoue@fc.ritsumei.ac.jp (D.I.); Noricobacter_pylori@softbank.ne.jp (N.N.); 3College of Pharmaceutical Sciences, Ritsumeikan University, 1-1-1 Noji-higashi, Kusatsu, Shiga 525-8577, Japan; 4Faculty of Pharmaceutical Sciences, Doshisha Women’s College of Liberal Arts, Kodo, Kyotanabe, Kyoto 610-0395, Japan; shkimura@dwc.doshisha.ac.jp; 5Department of Biopharmaceutics, Kyoto Pharmaceutical University, 5 Misasagi Nakauchi-cho, Yamashina-ku, Kyoto 607-8414, Japan; hkatsumi@mb.kyoto-phu.ac.jp (H.K.); yamamoto@mb.kyoto-phu.ac.jp (A.Y.)

**Keywords:** Caco-2, Calu-3, MDCK, three-dimensional mucociliary tissue model system, nasal drug absorption, apparent permeability coefficient

## Abstract

Recently, various types of cultured cells have been used to research the mechanisms of transport and metabolism of drugs. Although many studies using cultured cell systems have been published, a comparison of different cultured cell systems has never been reported. In this study, Caco-2, Calu-3, Madin–Darby canine kidney (MDCK), EpiAirway and MucilAir were used as popular in vitro cell culture systems, and the permeability of model compounds across these cell systems was evaluated to compare barrier characteristics and to clarify their usefulness as an estimation system for nasal drug absorption in rats. MDCK unexpectedly showed the best correlation (*r* = 0.949) with the fractional absorption (*F*_n_) in rats. Secondly, a high correlation was observed in Calu-3 (*r* = 0.898). Also, Caco-2 (*r* = 0.787) and MucilAir (*r* = 0.750) showed a relatively good correlation with *F*_n_. The correlation between *F*_n_ and permeability to EpiAirway was the poorest (*r* = 0.550). Because EpiAirway forms leakier tight junctions than other cell culture systems, the paracellular permeability was likely overestimated with this system. On the other hand, because MDCK formed such tight cellular junctions that compounds of paracellular model were less likely permeated, the paracellular permeability could be underestimated. Calu-3, Caco-2 and MucilAir form suitable cellular junctions and barriers, indicating that those cell systems enable the precise estimation of nasal drug absorption.

## 1. Introduction

In vitro studies using cultured cells have been carried out to predict drug absorption and to clarify the influence of metabolic enzymes and transporters associated with in vivo membrane transport of drugs after oral [1,2], nasal [3,4], pulmonary [5], and dermal [6] administration. Experiments using cultured cells are useful and effective in clarifying the mechanisms of the membrane permeation of drugs, because they can simply simulate in vivo situations and provide detailed information. Various cultured cell lines and primary cell culture models have been used to determine the nasal permeability of a drug. However, not all cultured cells are highly versatile. In particular, the easy and constant availability and the quality of supplied tissues could be problematic in the case of primary cell cultures from, for example, bovine, canine, ovine, and porcine animals.

Recently, some human respiratory and nasal epithelial cell culture systems, such as the EpiAirway and MucilAir systems, have been developed and used to estimate in vivo human nasal permeability. These systems are ready-to-use and feature three-dimensional mucociliary tissue models consisting of epithelial cells and goblet cells from the trachea/bronchus (EpiAirway) and the bronchus (MucilAir) of healthy humans. These commercial systems are also available as co-culture systems, which can be used for the estimation of not only human nasal permeability, but also toxicity [7] and virus infection [8], among others. Specifically, these systems adequately reflect the in vivo phenotypes of barriers, mucociliary responses, infection, toxicity responses, and disease. Thus, these systems are useful tools for nasal absorption studies and can be used in various types of experiments. However, cost and time lag are restrictions preventing the easy use of these culture systems. Especially, the manufacturer of EpiAirway indicates that the shelf life under storage at 4 °C, including the delivery time of tissues, may be up to 3 days, and that extended storage periods are not recommended unless necessary.

In contrast, cell lines such as Caco-2, MDCK (Madin–Darby canine kidney), and Calu-3, which are used in a large number of laboratories, are highly versatile for drug absorption studies. Caco-2 is a human adenocarcinoma cell line that has been widely used as an in vitro model of passive and carrier-mediated intestinal drug absorption [9,10]. Caco-2 cells differentiate into polarized monolayers with characteristics of the small intestine under normal culture conditions, although they originate from the colon. Previously, we used Caco-2 permeability to predict the rat nasal absorption of passively-permeated drugs, and clarified a good correlation between nasal absorption and Caco-2 permeability [11,12].

MDCK cells which grow and differentiate rapidly can be used for transport studies 3 days after seeding on filter inserts [13]. They are potential alternative models to mimic transport across the blood–brain barrier (BBB) because p-glycoprotein and tight junction proteins such as claudin-1, claudin-4 and occludin, which are important to form a restrictive paracellular barrier with tight junctions [14,15], are expressed. However, care should be taken to differentiate between nasal and other mucosa. Calu-3 is a mucus-producing submucosal gland carcinoma cell line derived from human bronchial epithelium that is capable of forming tight, polarized, and well differentiated cell monolayers [16]. Calu-3 cells also possess other functional characteristics such as asymmetric transferrin transport [17], the presence of functional cytochrome P450 isozymes (1A1, 2B6, and 2E1), and the expression of the cystic fibrosis transmembrane conductance regulator (CFTR) [18], similar to in vivo respiratory epithelia. Indeed, the permeability characteristics of Calu-3 cells correlate well with those of in vitro primary cultured rabbit tracheal epithelial cells (*r*^2^ = 0.91) and the rate of in vivo drug absorption from rat lung (*r*^2^ = 0.94), making this model potentially useful for prediction of absorption of molecules delivered through respiratory routes [19]. However, a systematic study on the correlation between in vitro Calu-3 permeability and in vivo nasal absorption has not been performed.

The aim of the present study was to clarify the characteristics of commercially available cell systems and cell lines by investigating the relationship between the permeability of compounds across each cultured cell system and the in vivo nasal absorption of rats, to clarify usefulness as an estimation system for nasal drug absorption. The model drugs were selected in terms of the *P*_app_ of drugs, where *P*_app_ values of 10^−7^ to 10^−5^ cm/s through a Caco-2 monolayer are commonly used to estimate the oral absorption of drugs. In this study, mannitol, inulin, sulfanilic acid, acyclovir and atenolol were selected as the paracellular transport model, while antipyrine, quinidine and methotrexate make up the transcellular model.

## 2. Materials and Methods

### 2.1. Materials

EpiAirway tissues (AIR-100) grown on polycarbonate inserts (0.6 cm^2^) were purchased from MatTek Corporation (Ashland, MA, USA). The assay medium (AIR-100-ASY) for treating and culturing EpiAirway was provided by the same company. MucilAir tissues grown on polycarbonate inserts (0.33 cm^2^) were supplied by Epithelix SàRL (Geneva, Switzerland). MucilAir culture medium (EP05MM) was obtained from the same company. Caco-2, MDCK, and Calu-3 cells were purchased from Dainippon Pharmaceuticals Co. (Osaka, Japan). Reagents and the media used for Caco-2, MDCK, and Calu-3 cultures and the preparation of monolayers were purchased from Sigma-Aldrich (St Louis, MO, USA) and Gibco Laboratories (Lenexa, KS, USA), respectively. Atenolol, antipyrine, acyclovir, inulin, methotrexate, quinidine, and sulfanilic acid were purchased from Sigma-Aldrich. [^3^H] inulin and [^14^C] D-mannitol for EpiAirway studies were purchased from PerkinElmer (Boston, MA, USA). [^14^C] inulin and [^14^C] D-mannitol for Caco-2 and Calu-3 studies were the product of American Radiolabeled Chemicals Inc. (St. Louis, MO, USA). All other chemicals were of reagent grade and are commercially available.

Eight-week-old male Wistar rats were purchased from Japan SLC Inc, (Shizuoka, Japan). All animal studies were previously approved by the Committee of the Animal Care of Shujitsu University and conducted under the Guideline (Approval ID: 013-002, 25 May 2015). Rats were housed under controlled temperatures at 20–26 °C and humidity at 40–60%, with a 12 h light/dark cycle, and were fed ad libitum until the day before in vivo experiments.

### 2.2. Culture of EpiAirway and MucilAir

Upon arrival, the primary human tracheal/bronchial epithelial cell system, EpiAirway, and the primary human bronchial epithelial cell system, MucilAir, were transferred to the well filled with 0.9 mL assay medium for EpiAirway and 0.7 mL MucilAir culture medium, which are serum free and contain growth factors and phenol red and are supplemented by default with antibiotics (penicillin/streptomycin). Both tissues were subsequently placed in an incubator with 5% CO_2_ at 37 °C. Both tissues were cultured in an air–liquid interface condition. EpiAirway tissues were used for the transport study following the overnight equilibration in assay medium. The mucous surface of MucilAir tissues was washed with 0.2 mL culture medium 2 days before the experiment in order to adjust the mucous level of each well. The permeation experiment using EpiAirway and MucilAir was conducted according to the manufacturer’s instructions.

### 2.3. Culture of Caco-2 and Preparation of Caco-2 Monolayers

Cells of the human colon adenocarcinoma cell line, Caco-2, were grown in Dulbecco’s Modified Eagle Medium (DMEM) supplemented with 10% fetal bovine serum (FBS), 1% L-glutamine, 1% non-essential amino acid, and 0.5% antibiotic–antimycotic solution [20]. Caco-2 cells were harvested with trypsin-EDTA and seeded on polycarbonate filters (pore size: 0.3 μm, growth area: 0.9 cm^2^, 12 wells/plate, Beckton Dickinson Bioscience, Bedford, MA, USA) at a density of 1.8 × 10^5^ cells/well. The culture medium was changed every 2 days. The monolayer was used for in vitro transport studies, 17–20 days after seeding.

### 2.4. Culture of Calu-3 and Preparation of Calu-3 Monolayers

Cells of the human bronchial submucosal gland carcinoma cell line, Calu-3, were grown in DMEM nutrient mixture F-12 HAM supplemented with 10% FBS and 0.5% antibiotic–antimycotic solution [21]. Calu-3 cells were harvested with trypsin-EDTA and seeded on polycarbonate filters (pore size: 0.3 μm, growth area: 0.9 cm^2^, 12 wells/plate, Beckton Dickinson Bioscience) at a density of 4.5 × 10^5^ cells/well. The culture medium was changed every 2 days. The monolayer was used for in vitro transport studies, 9–10 days after seeding.

### 2.5. Culture of MDCK and Preparation of MDCK Monolayers

The epithelial Mardi–Darby canine kidney cell line (MDCK) cells were grown in Modified α-Eagle Medium supplemented with 10% FBS and 0.5% antibiotic–antimycotic solution [13]. MDCK cells were harvested with trypsin-EDTA and seeded on polycarbonate filters (pore size: 0.3 μm, growth area: 0.9 cm^2^, 12 wells/plate, Beckton Dickinson Bioscience) at a density of 1.8 × 10^5^ cells/well. The culture medium was changed every 2 days. The monolayer was used for in vitro transport studies, 6–7 days after seeding.

### 2.6. Measurement of Transepithelial Electrical Resistance

The transepithelial electrical resistance (TEER) of cultured cell monolayers was measured with Millicell ERS-2 (Merck Millipore, Darmstadt, Germany) previous to the transport study. The values were normalized with the surface area and expressed as Ω·cm^2^. TEER is widely accepted as an index of the integrity of tight junctions. TEERs of each cell monolayers used in this study are shown in Figure 1.

### 2.7. In Vitro Transepithelial Transport Study

In the transport study using EpiAirway, all the procedures were based on the Drug Delivery Protocol established by MatTek Corporation (https://www.mattek.com/wp-content/uploads/EpiAirway-Drug-Delivery-Protocol.pdf), and carried out in Absorption Systems LP. Dulbecco’s Phosphate Buffered Saline (DPBS) without Ca^2+^ and Mg^2+^ (pH 7.4) was used as the assay buffer. Before the experiments, the tissues were washed twice with DPBS. The receiver and donor chambers were filled with either 0.75 or 0.3 mL of DPBS buffer at 37 °C. All test compounds were added to the apical compartment of the insert at a concentration of 10 µM. Inulin at 10 µM consisted of 1 µM of [^3^H] inulin and 9 µM of cold inulin. Mannitol at 10 µM consisted only of the radiolabeled compound. Radioactivity was 2 µCi/mL for the dosed [^3^H] inulin and 0.5 µCi/mL for the dosed [^14^C] mannitol. Thereafter, aliquots of the sample were taken from the basal side up to 60 min.

In transport studies using Caco-2, Calu-3, MDCK, and MucilAir, a transport medium (TM; Hank’s balanced salts solution supplemented with 15 mM glucose and 10 mM HEPES good buffer, pH 7.4) was used. All cell monolayers were preincubated with test compound-free TM at 37 °C for 10 min. The medium was replaced with TM (0.8 mL for Caco-2, Calu-3, and MDCK or 0.1 mL for MucilAir) containing compounds (10 µM) in the apical side and test compound-free TM (2.0 mL for Caco-2, Calu-3, and MDCK or 0.6 mL for MucilAir) in the basal side at 37 °C. Thereafter, aliquots of the sample were taken from the basal side up to 60 min.

The permeability (apparent permeability coefficient, *P*_app_ (cm/s)) of the drug was calculated according to the following equation:(1)$$P_{\mathrm{app}}=\frac{\;dQ/dt\;}{C_{0}\cdot A}$$
where *d**Q*/*d**t*, *C*_0_, and *A* are the appearance rate of drugs in the basal compartment (nmol/s), the initial drug concentration in the apical compartment (10 µM), and the surface area of the monolayer (0.9 cm^2^ for Caco-2, Calu-3 and MDCK, 0.6 cm^2^ for EpiAirway and 0.33 cm^2^ for MucilAir), respectively. The permeability of mannitol across MucilAir was cited from the literature [23].

### 2.8. In Vivo Study for Fractional Nasal Absorption of Atenolol, Antipyrine and Quinidine in Rats

The fractional absorption (*F*_n_) values of atenolol, antipyrine and quinidine following nasal application to rats were obtained using the procedures of intravenous bolus injection and nasal administration, which were reported in our previous study [11], as well as in additional experiments.

#### 2.8.1. Intravenous Bolus Injection

Under anesthesia by intraperitoneal sodium pentobarbital (50 mg/kg, Nembutal, Abbott Laboratories, Abbott Park, IL, USA), the right femoral artery was cannulated with polyethylene tubing (SP-31, Natsume, Tokyo, Japan) for the collection of blood samples. A drug dissolved in physiological saline (0.1 mL/kg B.W.) was injected into the left femoral vein. Blood samples were collected from arterial tubing in heparinized tubes at predetermined time intervals for 120 min. The blood was centrifuged to obtain the plasma.

#### 2.8.2. Nasal Administration

Under light isoflurane anesthesia, the right femoral artery was cannulated with polyethylene tubing. A drug dissolved in 5 µL of phosphate buffer saline (pH 7.4) was instilled at 1 cm depth from the nostril by microsyringe. Thereafter, animals were kept in a cage (KN-326-III, Natsume, Tokyo, Japan) throughout the experiment. It usually took 5 min after drug instillation for the rat to recover from the anesthesia. Blood samples were collected for 360 min after drug administration. During this period, the animal was allowed free access to water.

Fractional nasal absorption (*F*_n_) was calculated as follows:*F*_n_ = *AUC*_n_/*AUC*_iv_(2)
where *AUC*_n_ and *AUC*_iv_ are the area under the concentration–time profile following the nasal and intravenous administration of the drug. *AUC* was calculated according to the trapezoidal rule up to the last sampling point and extrapolation after the last sampling point.

### 2.9. Sample Assay

#### 2.9.1. Transport Study using Calu-3 and MDCK

Atenolol, antipyrine, acyclovir, methotrexate, quinidine, and sulfanilic acid present in the samples from Calu-3 and MDCK were analyzed by HPLC (LC-2010C HT, Shimadzu, Kyoto, Japan) equipped with a reverse-phase column (YMC-Pack Pro C18 RS, 150 × 4.6 mm, YMC Co. Ltd., Kyoto, Japan). The mobile phase was 20 mM sodium phosphate monobasic at a flow rate of 0.5 mL/min. The detection wavelengths were 254 nm for atenolol, antipyrine, acyclovir, quinidine, and sulfanilic acid, and 304 nm for methotrexate.

#### 2.9.2. Transport Study using EpiAirway

Atenolol, antipyrine, acyclovir, methotrexate, quinidine, and sulfanilic acid present in the samples from the EpiAirway transport study were analyzed with a liquid chromatography tandem mass spectrometry (LC-MS/MS) system (PE Sciex API 3000, AB Sciex, Framingham, MA, USA) equipped with reverse-phase columns. An Atlantis HILIC Silica column (3 μm, 50 × 2.1 mm i.d., Waters, Milford, MA, USA) was used for sulfanilic acid and a Keystone Hypersil BDS C18 column (3 μm, 30 × 2.0 mm i.d., Thermo Fischer Scientific, Waltham, MA, USA) for the other compounds. The mobile phase used for the separation of sulfanilic acid and other compounds was water and acetonitrile which contained 10% ammonium formate buffer (25 mM, pH 3.5), respectively. The separation of sulfanilic acid was performed with a gradient starting at 100% acetonitrile/0% water to 0% acetonitrile/100% water in 1.5 min and increasing to 100% water immediately, remaining in this condition for 2.0 min. The separation of atenolol, antipyrine, acyclovir, methotrexate and quinidine was performed with a gradient starting at 0% acetonitrile/100% water to 100% acetonitrile/0% water in 1.5 min, followed by isocratic separation for 0.5 min and increasing to 100% water immediately, remaining in this condition for 1.5 min. The flow rate and the injection volume were 0.3 mL/min and 30 μL for sulfanilic acid, 0.3 mL/min and 10 μL for the other five compounds. To determine [^3^H] inulin and [^14^C] d-mannitol, scintillation counting was performed for 5 min using a liquid scintillation counter, LSC3500 (Hitachi, Ltd., Tokyo, Japan).

#### 2.9.3. Transport Study using Mucilair and Animal Study

Atenolol, antipyrine, acyclovir, quinidine, and sulfanilic acid in the samples from transport studies with MucilAir were analyzed with a liquid chromatography and mass spectrometry (LC/MS) system (APL1100, Agilent Technology, Santa Clara, CA, USA) equipped with the reverse-phase column Luna C8 (5 μm, 250 × 4.6 mm i.d., Phenomenex Inc., Torrance, CA, USA) for sulfanilic acid and the Kinetex 2.6 μm C18 100 Å column (2.1 × 150 mm i.d., Phenomenex Inc.) for the other compounds. The mobile phases, flow rates, and injection volumes for the determination of the compounds were as follows:Atenolol    10 mM ammonium formate buffer/acetonitrile = 70:30, 0.2 mL/min, 15 μL
Acyclovir   10 mM ammonium formate buffer/methanol = 85:15, 0.1 mL/min, 50 μL
Antipyrine  5 mM ammonium formate buffer/acetonitrile = 70:30, 0.1 mL/min, 3 μL
Sulfanilic acid  0.1 mM ammonium formate buffer/acetonitrile = 75:25, 0.8 mL/min, 10 μL
Quinidine   0.2% formate/acetonitrile         = 70:30, 0.1 mL/min, 10 μL
Methotrexate  50 mM ammonium formate buffer/methanol  = 70:30, 0.2 mL/min, 10 μL

### 2.10. Data Fitting

The data except for methotrexate were fitted to Hill’s sigmoidal equation, shown below using a Kaleida Graph (Synergy Software Version 4.5, Reading, PA, USA):(3)$$F_{n}=\;\frac{Abs_{\max}\times P_{\mathrm{app}}{}^{\gamma }}{P_{\mathrm{app}\cdot 50}{}^{\gamma }\;+\;P_{\mathrm{app}}{}^{\gamma }}$$
where *F*_n_, *Abs*_max_, *P*_app·50_ and *γ* are the fractional absorption, maximum fractional absorption (100%), *P*_app_ required for 50% absorption and Hill coefficient, respectively.

## 3. Results

### 3.1. In Vitro Transepithelial Transport Study

The TEER of Caco-2, Calu-3, MDCK monolayers and MucilAir system was measured using Millicell ERS-2, while that of the EpiAirway system was determined in a previous study [23]. As listed in Table 1, the rank order of the TEER in the studied cell systems is MDCK (2120 ± 45 Ω·cm^2^) >> Caco-2 (771 ± 8 Ω·cm^2^) > MucilAir (560 ± 34 Ω·cm^2^) > Calu-3 (484 ± 9 Ω·cm^2^) > EpiAirway (391 ± 50 Ω·cm^2^).

Table 1 shows the *P*_app_ of eight model compounds to each cell monolayer and the fractional absorption (*F*_n_) following nasal application to rats. Some of them were reported in our previous study [11] and the others (atenolol, antipyrine and quinidine obtained) were determined by the same method. In this study, we assume that acyclovir, atenolol, inulin, mannitol, and sulfanilic acid permeate through the paracellular route, whereas the transcellular pathway is assumed for the other three compounds. As shown in Figure 2, the paracellular *P*_app_ of five compounds to EpiAirway was higher than those to the other monolayers. The *P*_app_ of the compounds to EpiAirway was similar (2.40 × 10^−6^–4.93 × 10^−6^ cm/s). This may be partly due to the characteristics of EpiAirway, which include the secretion of prosperous mucus and ciliated cells. A comparison of transcellular *P*_app_ of antipyrine, quinidine, and methotrexate to the cultured cell monolayers is shown in Figure 3. The *P*_app_ of antipyrine to all cell monolayers was similar and higher than 3.0 × 10^−5^ cm/s. The *P*_app_ of quinidine was higher than 1.0 × 10^−5^ cm/s, the rank order of which is Caco-2 > Calu-3 > MucilAir > EpiAirway > MDCK. Methotrexte showed the lowest *P*_app_ among all drugs.

### 3.2. Correlation of the Permeability with the Fraction Absorbed

Figure 4 shows the correlation between the fraction of compounds absorbed following nasal application to rats and the *P*_app_ of compounds, except for methotrexate, to EpiAirway (left) and MucilAir (right). The lines are the results of the fitting of the data to Hill’s sigmoidal equation. The Hill model has been widely used for the analysis on the data for which the range is limited. For example, the dose–response relationship is analyzed by the Hill equation, because the response of the drug is usually expressed as a percentage of the maximum potency (100%). According to a previous paper [11], the relationship between Caco-2 permeabilities and fractional nasal absorptions was well described by the Hill model. The correlation between *F*_n_ and *P*_app_ to EpiAirway was the poorest among all cell systems. No significant differences in the *P*_app_ of paracellular compounds (inulin, mannitol, sulfanilic acid, and acyclovir) to the EpiAirway system were observed. The *P*_app_ of six compounds (atenolol, antipyrine, acyclovir, mannitol, sulfanilic acid, and quinidine) to MucilAir was similar to those to MDCK, resulting in similar fitting curves for both systems.

Figure 5 shows the correlation between the fraction of compounds absorbed following nasal application to rats and the *P*_app_ of compounds, except for methotrexate, to Caco-2 (left), Calu-3 (middle), and MDCK (right). The correlation between the *P*_app_ to Caco-2 and *F*_n_ was described in our previous study [11]. However, in this study, the correlation of *P*_app_ to Caco-2 and *F*_n_ was re-analyzed, because the data for antipyrine, atenolol, and quinidine were added. The *P*_app_ for Caco-2 showed a relatively good correlation (*r* = 0.787) with *F*_n_. The correlation of *P*_app_ to Calu-3 and *F*_n_ (*r* = 0.898) was better than that observed in Caco-2. The cultured cell system indicating the best correlation (*r* = 0.949) with *F*_n_ was MDCK. According to Duff et al. [24], the molecular weight of inulin is so high that the permeation of inulin through the MDCK monolayer is very low, resulting in the difficulty in the evaluation of *P*_app_. The *P*_app_ values of paracellular model drugs to MucilAir were similar to these of MDCK. No other study reports the *P*_app_ of inulin to MucilAir, possibly for the same reason.

Methotrexate was expected to have a high *P*_app_ for cell monolayers, because, as Table 1 indicates, the *F*_n_ of methotrexate is high (98.2%). However, the *P*_app_ values of methotrexate to Caco-2, Calu-3, MDCK and MucilAir were quite low. According to some studies [25,26], methotrexate is transported to the basal side by the multidrug resistance-associated protein (MRP)3/Abcc3 on basolateral membrane. MRP3 is known to be expressed in the olfactory region of the nasal cavity of animals [27]. Furthermore, methotrexate is also a substrate of breast cancer resistance protein (BCRP)/ABCG2 [28] which is localized on the apical membrane of many cell lines [29]. Accordingly, higher in vivo nasal absorption may be due to the difference in the expression of MRP3 and BCRP in nasal epithelium and cultured cells. Although quinidine is a substrate of P-glycoprotein [30], which is an efflux transporter, quinidine exhibited higher permeability across each cell layer. On the other hand, the *P*_app_ of sulfanilic acid, which is diffused passively through the mucosal layer, to Caco-2 is shifted left from the sigmoid curve.

## 4. Discussion

Recently, a variety of cultured cells have been used for the research into the mechanism of the transport and metabolism and the pharmacological potency and safety of medicines and cosmetics. The experiment using the culture cell system is also important from the view point of the avoidance of animal sacrifice. Although many manuscripts using the cultured cell systems have been published, cultured cell systems, such as Caco-2, Calu-3 and MDCK, and human airway epithelium, such as EpiAirway and Mucilair systems, have never been compared. In this study, the passive transport of model compounds across these cell monolayers was evaluated to compare the barrier characteristics and to clarify their differences, with Caco-2 as a standard.

TEER is widely accepted as an index of tight junction integrity. The excised tracheal epithelia of various species, including human, have been reported to show TEER values of 100–300 Ω·cm^2^ [31,32,33]. TEERs of EpiAirway and MucilAir were similar to those of the excised tracheal epithelia. TEER of EpiAirway usually reaches 300–500 Ω·cm^2^. However, according to Chemuturi et al. [34], TEER of EpiAirway is less than 180 Ω·cm^2^. The discrepancy of results from some reports regarding TEER of EpiAirway might be due to the condition of the original tissue. The *P*_app_ of the paracellular model compounds to EpiAirway were higher (> 10^−6^ cm/s) as compared to the other cell systems, while the transcellular *P*_app_ to EpiAirway was similar with those to Caco-2 and MDCK. These findings indicate that the mechanism for the higher paracellular transport across EpiAirway is loose tight junctions. On the other hand, the *P*_app_ of paracellular model compounds to MDCK was the lowest among the cell models used in this study, because MDCK exhibits the highest TEER. According to Sauer et al. [7], MDCK expresses tight junction proteins such as Claudin-1, Claudin-4, and occludin, and forms a restrictive paracellular barrier with intercellular tight junctions, which is similar to Caco-2. Because the cellular junctions of MDCK are extremely tight, MDCK is sometimes used as the preferred model of the blood–brain barrier [35,36]. However, MDCK exhibited a *P*_app_ that is correlated well with *F*_n_ (Figure 5), because the *P*_app_ of inulin was excluded in the correlation of MDCK. The *P*_app_ to MDCK is likely to be too low to precisely estimate the nasal absorption of low-permeable compounds.

The mucous layer on the airway epithelia is composed of two distinctive gel-like layers: the superficial soluble mucous layer and the periciliary layer [37,38]. Because thick mucus layers covering the surface could reduce the mobility of molecules, generally, the superficial soluble mucous layer on the cell is an inhibitory factor for the transport of compounds, particularly those with high molecular weight or lipophilicity [39]. Calu-3 cells secrete mucus containing mucin [40]. However, the *P*_app_ of paracellular model compounds, particularly inulin (Mw: 5500), were similar to that to Caco-2 (no mucus secretion). Additionally, the *P*_app_ values of paracellular model compounds to EpiAirway were too high. These results suggest that the mucus on the cultured cell layer is not involved in the permeation of compounds. On the other hand, the *P*_app_ of paracellular model compounds to MucilAir were as low as those of MDCK, though the TEER of MucilAir was similar to those to Caco-2 and Calu-3. According to some manuscripts, mucin on the apical membrane does not affect the membrane permeation of small compounds [41]. The findings derived in this study failed to explain the low permeation of paracellular model compounds to MucilAir.

The *P*_app_ values of model compounds except for methotrexate to MucilAir were similar to those of MDCK monolayers, and exhibited a better sigmoidal correlation (*r* = 0.750) with *F*_n_ than that of EpiAirway (*r* = 0.550). No difference was observed among *P*_app_ values of the paracellular model compounds to EpiAirway, and the correlation coefficient between *P*_app_ and *F*_n_ was low, suggesting that EpiAirway cannot provide a good estimation of the paracellular permeability of compounds. MucilAir may allow the better estimation of nasal drug absorption in comparison to EpiAirway (Figure 4). However, because the paracellular permeation to MucilAir is likely underestimated, the error in the prediction of the nasal absorption of low permeable drug may be larger.

Methotrexate is a substrate of MRP3 [25,26] and BCRP [28]. MRP3 is localized in the basal membrane of human intestinal and lung-type cells [42] and can transport the substrate out of the cell, resulting in the inhibition of intracellular accumulation [43,44]. MRP3 is expressed in Caco-2 [29,45,46] and MDCK [47]. In Calu-3, the m-RNA of MRP3 was detected by reverse transcription polymerase chain reaction (RT-PCR) [29], whereas Sakamoto et al. [48] reported that the protein expression of MRP3 was negligible. Additionally, BCRP is also expressed in Caco-2 [29,45,46], Calu-3 [29,48], and MDCK [49]. The fractional absorption of methotrexate following application to the rat nasal cavity was high (98.2 ± 13.6%), although *P*_app_ values of methotrexate to Caco-2, Calu-3, MDCK and MucilAir were relatively low. *P*_app_ of methotrexate to only EpiAirway was high. This result might be due to the paracellular transport because of the low TEER and high drug permeation to EpiAirway. In order to estimate the nasal absorption with EpiAirway, it is necessary to investigate and/or clarify some other functions of EpiAirway. On the other hand, the *P*_app_ of methotrexate to MucilAir is plotted close to the sigmoid curve in Figure 4, although the *P*_app_ to MucilAir is as low as those to Caco-2, Calu-3 and MDCK. The protein expression or the activity of MRP3 and/or BCRP in MucilAir might be more suitable for the precise estimation of the nasal drug absorption as compared to other cell systems, while no manuscript has been published on the MRP3 and BCRP expression in MucilAir. The information on other transporters, in addition to MRP3 and BCRP, should be obtained for more precise estimation.

Quinidine is a substrate of P-glycoprotein [30]. Although P-glycoprotein is an efflux transporter, quinidine exhibited higher permeability across each cell layer. Quinidine has also been known as a substrate of organic cation transporters (OATs) and organic anion-transporting polypeptides (OATPs) [50], which can enhance the transepithelial quinidine transport. The passive permeability of quinidine across several cell layers is relatively high at high apical concentration [34]. Several uptake transporters such as OATs and OATPs can be involved in the absorptive transport of quinidine. That is why quinidine was used as a transcellular drug for the fitting analysis on *F*_n_. The *P*_app_ of sulfanilic acid to Caco-2 is shifted left from the sigmoid curve in Figure 5. The *P*_app_ of sulfanilic acid to Caco-2 is relatively low compared to those of other drugs. One of the possible mechanisms is p-glycoprotein, because the expression of p-glycoprotein in Caco-2 is higher than those of MucilAir and Calu-3 [46,48]. However, it is generally accepted that sulfanilic acid is diffused passively through the mucosal layer. Therefore, the reason for the low *P*_app_ of sulfanilic acid to Caco-2 is not clear.

In this study, the characteristics of five cultured cell systems used for estimation of fractional nasal absorption in rats were compared. The results clarified that the fractional absorption of drugs transported passively in rats can be predicted with some accuracy from *P*_app_ in Calu-3 or MucilAir. The estimation of the nasal absorption in human is difficult at present, because of the lack of data on the human nasal absorption of model drugs used in this study. Although the nasal cavity of humans and rats are different anatomically, the mucociliary clearance, which can affect the residence time and the absorption rate of the drug, is similar in humans and rats [12]. These functions are important determinants of the fractional nasal absorption of drugs. An estimation system using these cell systems can be developed in the future when more data on the nasal drug absorption in human is available.

## 5. Conclusions

In this study, the characteristics of five cultured cell systems which can be used for the estimation of nasal absorption were compared. Each compound is permeated across the cell layers through the paracellular and/or transcellular routes and by passive or active transport. The paracellular transport of model drugs across EpiAirway is too high, while that across MDCK is too low, as compared to the nasal mucosa of the rat. On the contrary, the barrier function of Caco-2, Calu-3 and MucilAir is likely similar to that of the rat nasal mucosa. Therefore, these cell systems could be a suitable system for the estimation of nasal absorption in rats. Although the cultured cell lines can be reasonably applicable to estimate the passive permeability of a compound, cell systems should be carefully selected according to the purposes of the research, paying attention to the integrity of the tight junction and the expression of transporters of cell systems.

## Figures and Tables

**Figure 1 pharmaceutics-12-00079-f001:**
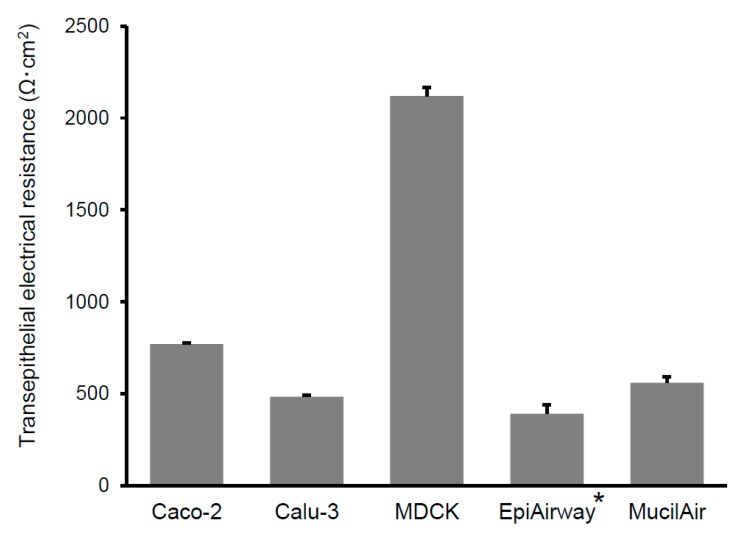
Comparison of transepithelial electrical resistance (TEER) of each cultured cell systems. The TEER of cultured cell systems was measured by Millicell-ERS (MILLIPORE) before the transport experiment. Bars represent the mean ± S.E. of 3–4 experiments. *: TEER value of EpiAirway was cited from the previous report [22].

**Figure 2 pharmaceutics-12-00079-f002:**
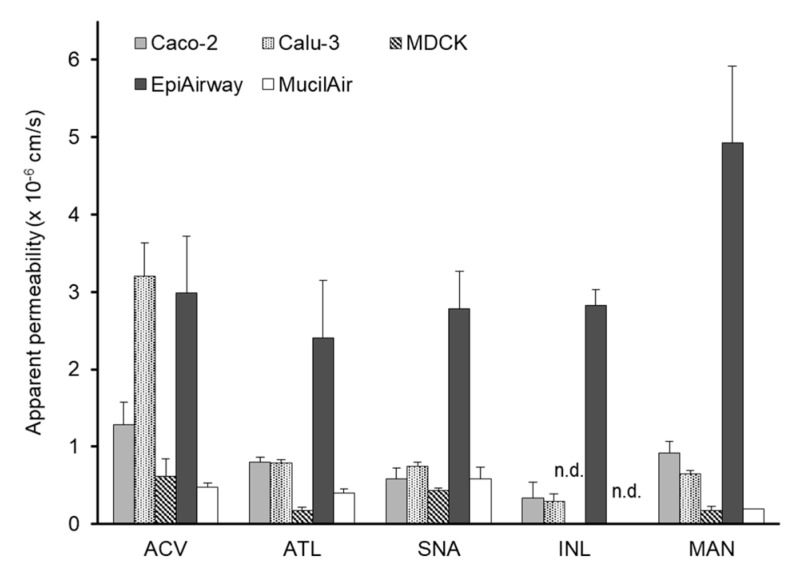
Comparison of apparent permeability of paracellular compounds across cultured cell systems. Keys: ACV: acyclovir, ATL: atenolol, SNA: sulfanilic acid, INL: inulin, MAN: mannitol, n.d.: not determined. Bars represent the mean ± S.E. of 3–4 experiments.

**Figure 3 pharmaceutics-12-00079-f003:**
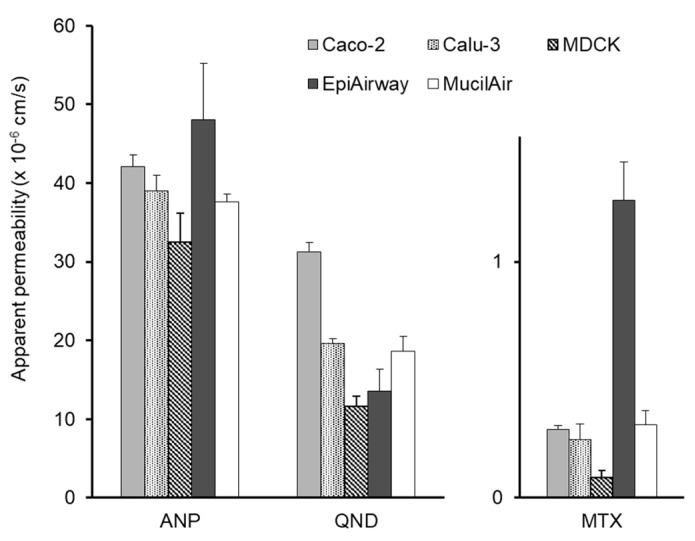
Comparison of apparent permeability of transcellular compounds across cultured cell systems. Keys: ANP: antipyrine, QND: quinidine, MTX: methotrexate. Bars represent the mean ± S.E. of 3–4 experiments.

**Figure 4 pharmaceutics-12-00079-f004:**
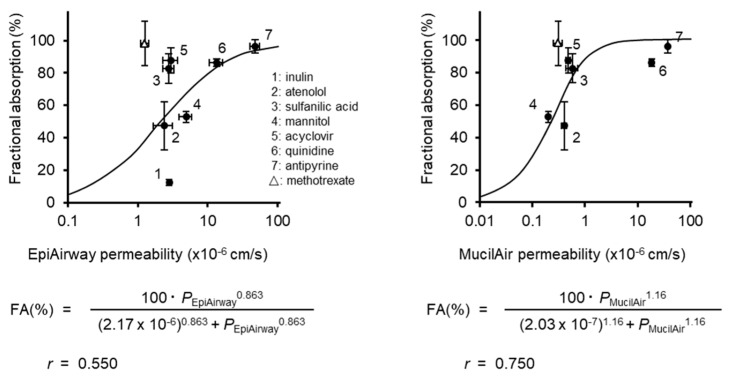
Correlation of the fractional absorption of compounds following nasal application to rats with permeability across EpiAirway (**left**) and MucilAir (**right**). Keys; 1: inulin, 2: atenolol, 3: sulfanilic acid, 4: mannitol, 5: acyclovir, 6: quinidine, 7: antipyrine, FA: fractional absorption. The data were fitted to Hill’s sigmoidal equation, with the exception of methotrexate. Fractional absorptions of compounds (except for antipyrine, atenolol and quinidine) were cited from Furubayashi et al. [11]. Data for methotrexate are plotted as an open triangle. Data are expressed as the mean ± S.E. of 3–4 experiments.

**Figure 5 pharmaceutics-12-00079-f005:**
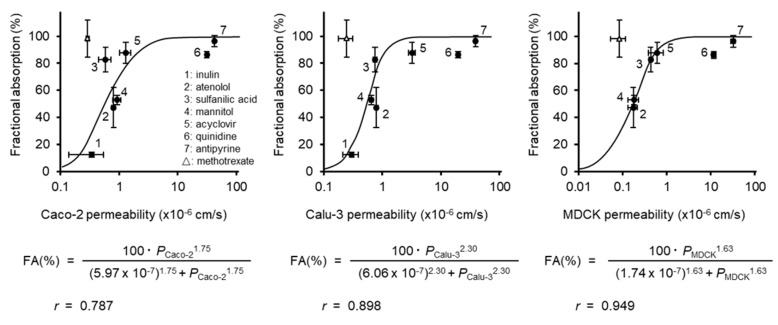
Correlation of the fractional absorption of compounds following nasal application to rats with respect to the permeability across Caco-2 (**left**), Calu-3 (**middle**) and MDCK (**right**) monolayer. Keys: 1: inulin, 2: atenolol, 3: sulfanilic acid, 4: mannitol, 5: acyclovir, 6: quinidine, 7: antipyrine, FA: fractional absorption. The data were fitted to Hill’s sigmoidal equation, with the exception of methotrexate. Fractional absorptions of compounds (except for antipyrine, atenolol and quinidine) were cited from Furubayashi et al. [11]. Data for methotrexate are plotted as an open triangle. Data are expressed as the mean ± S.E. of 3–4 experiments.

**Table 1 pharmaceutics-12-00079-t001:** Permeability across cultured cell systems and fractional nasal absorption of compounds.

Compounds	Permeability (×10^−6^cm/s ± S.E.)					Fractional Absorption*^a^ (% ± S.E.)
	Caco-2	Calu-3	MDCK	EpiAirway	MucilAir	
Acyclovir	1.29 ± 0.28	3.20 ± 0.43	0.61 ± 0.23	2.99 ± 0.73	0.48 ± 0.05	87.7 ± 7.9
Atenolol	0.80 ± 0.06	0.79 ± 0.05	0.17 ± 0.04	2.40 ± 0.74	0.40 ± 0.05	47.3 ± 14.9
Sulfanilic acid	0.58 ± 0.14	0.75 ± 0.05	0.43 ± 0.04	2.78 ± 0.40	0.58 ± 0.15	82.7 ± 9.0
Inulin	0.34 ± 0.20 *^1^	0.30 ± 0.09 *^1^	-	2.83 ± 0.20 *^2^	-	12.4 ± 2.0 *^1^
Mannitol	0.92 ± 0.15 *^3^	0.64 ± 0.04 *^3^	0.18 ± 0.05 *^3,b^	4.93 ± 0.99 *^3^	0.2 ± 0.0 *^4,c^	52.9 ± 3.4 *^3^
Antipyrine	42.1 ± 1.47	39.0 ± 2.02	32.5 ± 3.71	48.1 ± 7.15	37.6 ± 0.99	96.3 ± 4.2
Quinidine	31.2 ± 1.25	19.6 ± 0.60	11.6 ± 1.31	13.5 ± 2.86	18.6 ± 1.93	86.3 ± 2.4
Methotrexate	0.29 ± 0.02	0.25 ± 0.07	0.09 ± 0.03	1.26 ± 0.16	0.31 ± 0.06	98.2 ± 13.6

Data are expressed as the mean with S.E. of 3–4 experiments. *a: cited from our previous report [11,12]. *b: cited from the previous report [15]. *c: cited from the previous report [23]. *1: [^14^C] inulin, *2: [^3^H] inulin, *3: [^14^C] d-mannitol, *4: [^3^H] mannitol, -: not determined.

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
