# Peer review of "Comparison of Various Cell Lines and Three-Dimensional Mucociliary Tissue Model Systems to Estimate Drug Permeability Using an In Vitro Transport Study to Predict Nasal Drug Absorption in Rats"

_pharmaceutics, 2020, doi:10.3390/pharmaceutics12010079_

Round 1
Reviewer 1 Report
This paper is about the comparison of different types of monolayer or other 3D systems which are used as methods to evaluate the permeability of drugs after nasal adminstration. The paper is well written, interseting and worth of publication.
The only comment is about the fitting method. A separate section in the methods part should be added to precicely describe the method used.
Reviewer 2 Report
Tomoyuki Furubayashi et. al. have compared various in vitro cell culture systems such as Caco-2, Calu-3, MDCK, EpiAirway and MucilAir for the permeability of model drugs and have correlated this with nasal absorption of rats for the same drugs. The correlation of in vitro cell culture models and fractional nasal absorption in rats for model drugs has a novelty feature to it. But some of the in vitro as well as fractional nasal absorption studies for some drugs such as atenolol, antipyrine and quinidine reported in this paper has already been published by the same author. This has been addressed by the authors in the manuscript. This paper reports additional comparative studies with few more model drugs and correlation with nasal absorption. While interesting studies have been reported in this manuscript, several points needs to be addressed which are as follows.
Title:
Is not very clear, especially the part on “Nasal absorption of Rats.” Does not give the idea about in vivo studies reported in the manuscript. Could be reworded for better understanding.
Abstract:
Very comprehensive. Line 30 is not clear.
Introduction:
Line 94 is not very clear. No information on which drugs used and why those drugs were chosen to assess. Also, mentioning of which drugs are likely transported paracellularly and transcellularly will be helpful.
Materials and methods:
In 2.2 culture media or assay medium used is not mentioned.
In 2.7, methods for testing the drug permeability needs to be explained in detail. TEER values obtained for various cell lines if represented graphically would help in better understanding. In vivo methods for determining nasal absorption in rats needs to be elaborated.
In 2.9.2, line 205 manufacturer’s name for liquid scintillation counter used is not mentioned.
In 2.9.3, were inulin and mannitol assessed too?
Results:
Could be reworded for better understanding. In figure 1, reason for not determining permeability of inulin with MDCK and MucilAir is not mentioned.
Discussion:
Is a bit difficult to follow. Could be organized in respect to comparative studies and reworded for better understanding.
Conclusion:
The conclusion is vague.
Reviewer 3 Report
In this study, the authors investigated the correlation between in vivo nasal absorption and in vitro cell line systems and found some interesting results. I would like to recommend this paper for publication, despite some minor comments.
Regarding the correlation statistics, what is the rationale behind using Hill-equation? Sometimes, the R2 for model fitting does not necessary they are correlated. The authors need to justify this. In addition, there is an obvious outlier in figure 4. It is not easy to see which one it is, but I guess it is sulfanilic acid. In contrast, MucilAir produced better fitting for this drug. The authors need to discuss this point. The in vivo nasal absorption was conducted in the rat. The difference between humans and rats needs to be discussed. Minor suggestion: please put the legends for the markers for Figures 3 and 4. It is not clear in the current version.Author Response
Please see the attachment.

Round 2
Reviewer 2 Report
The authors have addressed all of the concerns that I raised previously. I commend them on this effort. I now only have a single concern, but it is of major concern. The quality of the English language writing in the manuscript remains poor. The manuscript should be thoroughly proofread and copy edited prior to acceptance.